# Investigation of the Phase Transitions and Magneto-Electric Response in the 0.9(PbFe$_{0.5}$Nb$_{0.5}$)O$_3$-0.1Co$_{0.6}$Zn$_{0.4}$Fe$_{1.7}$Mn$_{0.3}$O$_4$ Particulate Composite

**Krishnamayee Bhoi** [1], **Smaranika Dash** [1], **Sita Dugu** [2], **Dhiren K. Pradhan** [3], **Anil K. Singh** [1], **Prakash N. Vishwakarma** [1], **Ram S. Katiyar** [2] **and Dillip K. Pradhan** [1,*]

1 Department of Physics and Astronomy, National Institute of Technology, Rourkela 769008, Odisha, India; 515ph6004@nitrkl.ac.in (K.B.); 514ph1004@nitrkl.ac.in (S.D.); singhanil@nitrkl.ac.in (A.K.S.); prakashn@nitrkl.ac.in (P.N.V.)
2 Department of Physics, University of Puerto-Rico, P. O. Box 70377, San Juan, PR 00936-8377, USA; sita12318@gmail.com (S.D.); ram.katiyar@upr.edu (R.S.K.)
3 Extreme Materials Initiative, Geophysical Laboratory, Carnegie Institution for Science, Washington, DC 20015, USA; dhirenkumarp@gmail.com
* Correspondence: dillippradhan@nitrkl.ac.in

**Abstract:** Multiferroic composites with enhanced magneto-electric coefficient are suitable candidates for various multifunctional devices. Here, we chose a particulate composite, which is the combination of multiferroic (PbFe$_{0.5}$Nb$_{0.5}$O$_3$, PFN) as matrix and magnetostrictive (Co$_{0.6}$Zn$_{0.4}$Fe$_{1.7}$Mn$_{0.3}$O$_4$, CZFMO) material as the dispersive phase. The X-ray diffraction analysis confirmed the formation of the composite having both perovskite PFN and magnetostrictive CZFMO phases. The scanning electron micrograph (SEM) showed dispersion of the CZFMO phase in the matrix of the PFN phase. The temperature-dependent magnetization curves suggested the transition arising due to PFN and CZFMO phase. The temperature-dependent dielectric study revealed a second-order ferroelectric to the paraelectric phase transition of the PFN phase in the composite with a small change in the transition temperature as compared to pure PFN. The magnetocapacitance (MC%) and magnetoimpedance (MI%) values (obtained from the magneto-dielectric study at room temperature (RT)) at 10 kHz were found to be 0.18% and 0.17% respectively. The intrinsic *magneto*-electric coupling value for this composite was calculated to be 0.14 mVcm$^{-1}$Oe$^{-1}$, which is comparable to other typical multiferroic composites in bulk form. The composite PFN-CZFMO exhibited a converse magneto-electric effect with a change in remanent magnetization value of −58.34% after electrical poling of the material. The obtained outcomes from the present study may be utilized in the understanding and development of new technologies of this composite for spintronics applications.

**Keywords:** multiferroic; composites; phase transition; magneto-electric effect

## 1. Introduction

Multiferroics materials are characterized by the presence of more than one primary ferroic property such as ferroelectric, ferromagnetic, and ferroelastics in a single phase. The magneto-electric (ME) multiferroics material, which enables the cross-interaction between the ferroelectric and ferromagnetic order parameters, is termed ME coupling in multiferroics. This effect provides opportunities for possible applications in the miniaturization of multifunctional devices. These magnetoelectric multiferroics are promising candidates in the present R&D activity due to their technological applications as well as basic scientific significance. Generally, multiferroics materials are available in single-phase and composite structures. Unfortunately, most of the single-phase multiferroics show multiferroic behavior either at low temperature or the ME coupling is somewhat lower than that of composite structure at room temperature (RT). In contrast, two-phase ME multiferroic composites show a strong ME effect due to the interface interaction of the

constituent phases. This effect is the so-called "$0 + 0 \rightarrow 1$" product effect, which makes them more attractive [1]. Here neither of the constituent phases possess the ME effect but the ME effect observed in composites is a result of the product of magnetostrictive effect (magnetic/mechanical) from the magnetic phase and piezoelectric (mechanical/electrical) effect at the piezoelectric phase. Two-phase ME multiferroic materials in bulk form are applicable in AC/DC magnetic field sensors, tunable transformers, energy harvesters, memristors, and gyrators, etc. [2]. The two-phase composite can be described by the common connectivity scheme such as 0-3, 2-2, 1-3 type, etc., where each number signifies the connectivity of each phase. For example, the 0-3 composite describes the embedment of one phase (represented by 0) in the matrix of another phase (represented by 3). The commonly studied connectivity scheme for composite systems are 0-3–, 3-3–, 2-2–, 1-3–type structures of the ferroelectric and magnetic phase [3]. Out of these available connectivity schemes, the particulate (0-3–type) composite supports easier fabrications along with the control over the physical properties by changing the volume fractions of the individual phases [4]. So, in this report, a 0-3 –type connectivity scheme was chosen to study various properties of the prepared particulate multiferroic composite.

The idea behind designing multiferroic composites is to achieve good multiferroic behavior as well as a strong ME coupling coefficient over a wide range of operational temperatures. This initiated with a combination of suitable constituent phases for the composite formation, which includes the selection of one phase, which is dispersed in the matrix of the other phase (for particulate composite). Ahlawat et al. prepared $0.65Pb(Mg_{1/3}Nb_{2/3})O_3-0.35PbTiO_3$ (PMN-PT)/$NiFe_2O_4$ (NFO) nanocomposites using the sol-gel method. They studied the structural phase transition and the effect of electrical-poling on the ME behavior of the PMN-PT/NFO composite [5]. Sakanas et al. studied the dielectric and phonon spectroscopy of $(1-x)Pb_{0.988}(Zr_{0.52}Ti_{0.48})_{0.976}Nb_{0.024}O_3-xCoFe_2O_4$ [$(1-x)$PZTN–$x$CF] composites, prepared by the in situ sol-gel combustion method [6]. The composite of $0.9\ BaTiO_3-0.1\ Ni_xZn_{1-x}Fe_2O_4$ was fabricated by Upadhyaya et al. and investigated its magnetic, ferroelectric, and ME effects [7]. Yao et al. synthesized the composite of $(1-x)Ni_{0.4}Zn_{0.6}Fe_2O_4 + xPb(Zr_{0.53}Ti_{0.47})O_3$ using the solid-state reaction route. They reported the change in permittivity and permeability by the application of a magnetic and electric field [8]. The composite of $(x)Ni_{0.75}Zn_{0.25}Fe_2O_4-(1-x)\ BaTi_{0.85}Zr_{0.15}O_3$ was prepared by Adhalakha et al. using the solid-state reaction method and studying its structural, dielectric, and magnetic behavior [9]. Walther et al. observed hysteric ME behavior in $(CoFe_2O_4)x–(BaTiO_3)1-x$ synthesized using the polyol mediated synthesis route [10]. The $(1-x)BiFeO_3–xNaNbO_3$ nanocomposite was prepared by Ummer et al. using the pechini method and investigated the electric, magnetic, and optical properties [11]. Dwivedi et al. studied the low temperature magnetic and transport properties of $La_{0.7}Sr_{0.3}MnO_3– PbZr_{0.52}Ti_{0.48}O_3$ (LSMO–PZT), which was prepared using the sol-gel method [12]. Stingaciu et al. investigated the temperature-dependent magneto-dielectric behavior in $BaTiO_3–SrFe_{12}O_{19}$ nanocomposites fabricated using the spark plasma sintering method [13]. The enhanced magnetization value in $BiFeO_3–La_{0.7}Sr_{0.3}MnO_3$ composites due to the co-existence of multiple magnetic ordering present in the compound was reported by Pillai et al. [14]. Guerra et al. studied the ferroelectric, magnetic, and magneto-electric properties of the $(PbZr_{0.65}Ti_{0.35}O_3/BaFe_{12}O_{19})$ composite. They found the existence of a multiferroic behavior along with magnetoelectric behavior in this composite [15]. Mokhtari et al. performed a systematic study on the magneto-electric voltage coefficient of the $0.94Pb(Fe_{1/2}Nb_{1/2})O_3–0.06PbTiO_3/(Co, Ni)Fe_2O_4$ particulate composite and reported that the magnet-electric voltage coefficient strongly depended on the magnetic field dependence of the magnetostriction of the piezomagnetic phase [16]. Vertsioti et al. prepared the composites of $Pb(Zr_{0.52}Ti_{0.48})O_3– 5\%Fe_3O_4$ and observed the changes in the strain-electric field curves and the associated piezoelectric coefficient by the application of the external magnetic field [17]. Shi et al. prepared the particulate composite of $NiFe_2O_4-BaTiO_3$ by combining hydrothermal and sol-gel methods. They found a giant magneto-electric effect at the high frequency studied through both static and dynamic measurement [18].

Saha et al. prepared the particulate composite by the controlled precipitation of the garnet phase using additive ($MnO_2$) assisted sintering of $BiFeO_3$-$PbTiO_3$-$DyFeO_3$ pseudo ternary alloy and observed a 50% increase in the saturation polarization by the application of a magnetic field of 1 T [19]. Momin et al. fabricated the composite of $x$Li$_{0.1}$Ni$_{0.2}$Mn$_{0.6}$Fe$_{2.1}$O$_4$-$(1-x)$Bi$_{0.8}$Y$_{0.2}$FeO$_3$ (LNFMO-BYFO) and studied their structural, morphological, electrical, and magnetoelectric properties. They reported a maximum magneto-electric voltage co-efficient of $182 \times 10^3$ $Vm^{-1}T^{-1}$ for the 0.1LNFMO-0.9BYFO composite [20]. Samghabadi et al. obtained the highest magneto-electric coupling coefficient of 4.3 mV/cm Oe in the 0.2CoFe$_2$O$_4$-0.8BaTiO$_3$ particulate composite [21]. The above literature survey on multi-ferroic composite revealed that multiferroic composites are fabricated by combining (a) ferroelectric compounds as a matrix and ferromagnetic/antiferromagnetic/ferromagnetic as dispersive phase, (b) distribution of ferromagnetic/antiferromagnetic/ferrimagnetic phase in the matrix of multiferroic phase, (c) multiferroic and ferroelectric phases, and (d) multiferroic phases in order to achieve improved ME behavior. In the present study, we chose to prepare the composites of Pb(Fe$_{0.5}$Nb$_{0.5}$)O$_3$ (PFN) and Co$_{0.6}$Zn$_{0.4}$Fe$_{1.7}$Mn$_{0.3}$O$_4$ (CZFMO). The perovskite oxide PFN is one of the multiferroic candidates used for ceramic capacitors, piezoelectrics, electrostatics, pyroelectrics, etc. [22]. It is a multiferroic system where ferroelectric properties arise from the ordering of Pb$^{2+}$ in site A, Nb$^{5+}$ in B″-site, and magnetic characteristics from Fe$^{3+}$ in the B′-site of the A′A″(B′B″)O$_3$ perovskite structure [23]. It undergoes a ferroelectric-to-paraelectric phase transition around 380 K and G-type antiferromagnetic ordering below 140 K. Further, PFN can be sintered at a relatively low temperature (1373 K) as compared to other conventional ceramics such as BaTiO$_3$, CaTiO$_3$ (sintering temp = 1573 K) [24]. Among the available magnetostrictive ferrite, CoFe$_2$O$_4$ (CFO) possesses the high value of the absolute magnetostriction coefficient $\lambda = -110 \times 10^{-6}$ with a saturation magnetization value of M$_S$ = 81 emu/g [25]. However, a small ME coefficient (5-6 mV/cm-Oe) was reported for the PbZr$_{0.52}$Ti$_{0.48}$O$_3$-CoFe$_2$O$_4$ (PZT-CFO) particulate composite [26]. This is due to the large magnetic anisotropy and coercivity of CFO, which obstructs the domain rotation and domain wall motion. The Mn-modified CFO, i.e., CoFe$_{1.7}$Mn$_{0.3}$O$_4$, is found to minimize the anisotropy constant and coercivity. Additionally, the Zn-modified CFO reported a decrease in the magnetic anisotropy in CFO [27]. The optimum composition of the Zn- and Mn-modified CFO, Co$_{0.6}$Zn$_{0.4}$Fe$_{1.7}$Mn$_{0.3}$O$_4$ (CZFMO), shows a maximum in-plane piezomagnetic coefficient of $|q| \approx 0.070$ ppm/Oe, with sizeable M$_S$ and longitudinal and transverse magnetostriction coefficients of $\lambda_{11} \approx -20$ ppm and $\lambda_{12} \approx -10$ ppm [27]. Further, CZFMO has high resistivity ($10^8 \Omega$ cm), which restricts the piezoelectric charge leakage, consequently, satisfies the essential criterion mention by Boomgaard et al. [28]. In our previous report, we prepared a series of (1-Φ) PbFe$_{0.5}$Nb$_{0.5}$O$_3$-ΦCo$_{0.6}$Zn$_{0.4}$Fe$_{1.7}$Mn$_{0.3}$O$_4$ composites and studied the structural, ferroelectric, magnetic, magneto-electric properties at RT [29]. In this report, we studied the temperature-dependent magnetic, dielectric properties (to study the phase transitions behavior) along with the magneto-dielectric properties at RT of the 90wt.% PFN-10wt.% CZFMO composite (abbreviated as PC10 throughout the manuscript).

## 2. Experimental Details

The multiferroic PC10 composite was prepared using a hybrid synthesis technique. The details of the synthesis condition of (1-Φ)PFN-ΦCZFMO composite were described in our previous report [29]. The structural analysis was carried out by X-ray diffraction technique using D-PHASER (Bruker) X-ray diffractometer with CuK$_{\alpha1}$ radiation of $\lambda$ = 1.5405 Å. The XRD data were collected in a wide range of Bragg's angle 2θ ($20° \leq 2\theta \leq 70°$), with the step size of 0.02° at a scan speed of 1 degree/min. The surface morphological features were studied using an automated Scanning Electron Microscope (SEM-JEOL-JSM 6480 LV, JEOL, Tokyo, Japan). The sintered pellet was gold-coated under vacuum prior to the SEM investigations. The SEM images were captured at various magnifications. The magnetic properties of the composite, such as magnetization versus temperature (M-T) in zero-field cooling and field cooling conditions and magnetization versus magnetic field (M-H) at

various temperatures, were measured using DynaCool physical properties measurement system (PPMS) from Quantum Design (San Diego, CA, USA). Temperature-dependent dielectric properties were recorded over a wide range of frequencies and temperatures with the help of the HIOKI IM3570 impedance analyzer, Japan coupled with a laboratory-made sample holder and temperature controller. The RT magneto-dielectric (MD) properties were measured using a high precession impedance analyzer (Wayne Kerr 6500B, West Sussex, UK) coupled with an electromagnet (GMW-5403) attached with a bipolar dc supply. The converse magneto-electric effect was probed by measuring the M-H data before and after electric field poling using VSM (Lakeshore USA 7404) in the field range of −1.5 T to 1.5 T at RT. The samples were poled at 20 kV/cm at RT.

## 3. Results & Discussion

### 3.1. Structural and Morphological Study

Figure 1a shows the room temperature X-ray diffraction pattern of the sintered PC10 composite. It is clear that the two sets of peaks corresponding to the perovskite PFN phase (designated as 1) and spinel CZFMO phase (designated as 2) were observed in the diffraction pattern. The obtained distinct diffraction peaks were indexed to the monoclinic structure of a pure PFN phase with space group *Cm* and cubic structure of CZFMO with space group *Fd-3m* [30–32]. The XRD pattern showed the coexistence of both the multiferroic PFN and ferrite CZFMO phase without an appearance of any other secondary phases (within the resolution limit of the equipment), and hence, suggests the formation of composites.

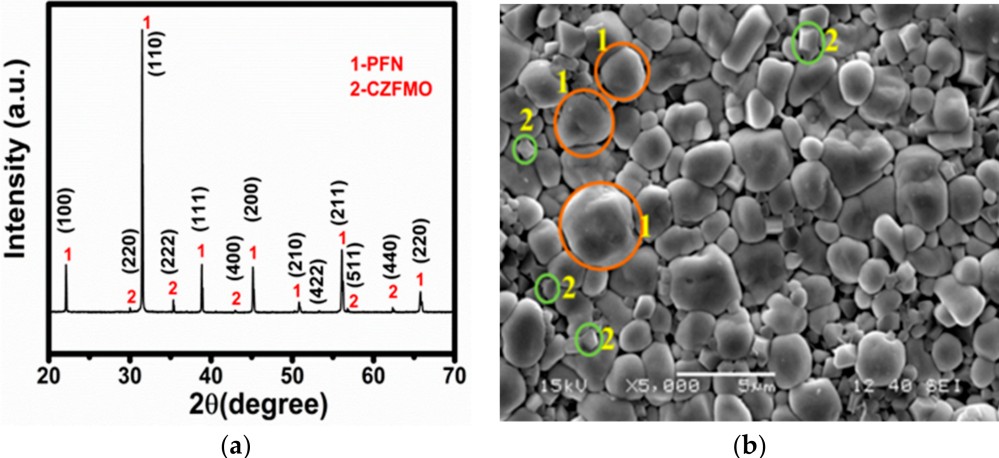

(**a**)                                          (**b**)

**Figure 1.** (**a**) Room temperature X-ray diffraction pattern and (**b**) SEM micrograph of PC10 composite.

In Figure 1b, the SEM micrograph shows the composite with larger grain attributed to the PFN (referred to as one) phase while smaller grain corresponds to the CZFMO (referred to as two) phase. The grain size of the PFN phase was in the micrometer range, whereas the grain size of the CZFMO phase was in the nanometer range. In the micrograph, one could see the CZFMO phase (phase 2) embedded in the matrix of the PFN phase (phase 1), which revealed a successful formation of the particulate composite without any secondary phases. This is in accordance with the observed XRD data. So the preliminary structural and microstructural analysis proposed the formation of the high-quality particulate composite and indicated the absence of any interdiffusion between the constituent phases that may lead to secondary phases.

### 3.2. Magnetic Study

The temperature-dependent magnetization (M-T) curves were measured in the temperature range 2 K–400 K in zero-field cooling (ZFC) and field cooling (FC) conditions at two different magnetic fields 500 Oe and 1000 Oe (shown in Figure 2). The ZFC and FC

curves overlapped in the high-temperature region for both the magnetic fields. As the low temperature was approached, large irreversibility and branching of ZFC and FC curves for both the fields arose, suggesting the onset of blocked/hindered spins. The overlapping temperature was $T_{overlap}{\sim}330$ K in the presence of a magnetic field H = 500 Oe and shifted towards a lower temperature $T_{overlap}{\sim}300$ K for H = 1000 Oe. Since this type of feature is generally observed for spin glass materials and has also been reported for CZFMO [33], it helped us to recognize it to be due to the CZFMO phase in the present composite. Earlier reports on the M-T curve of PFN suggested an antiferromagnetic to paramagnetic phase transition in the range of 140 K–158 K for bulk ceramics (170 K–200 K in thin films) [34,35]. In the M-T graph, a cusp was seen around 135 K (referred to as $T_N$) that could be assigned as antiferromagnetic to paramagnetic phase transition. These characteristics in this composite could be attributed to the PFN phase. On further increasing the temperature, another peak was observed around 393 K, which was close to the ferroelectric to the paraelectric phase transition ($T_C$) for the PFN phase. This could be considered as an indication of a strong ME-coupling behavior present in the system. However, this transition was at the end of the measuring temperature range, so measurement in a wider range may be required to certify the observed $T_C$ in M-T data.

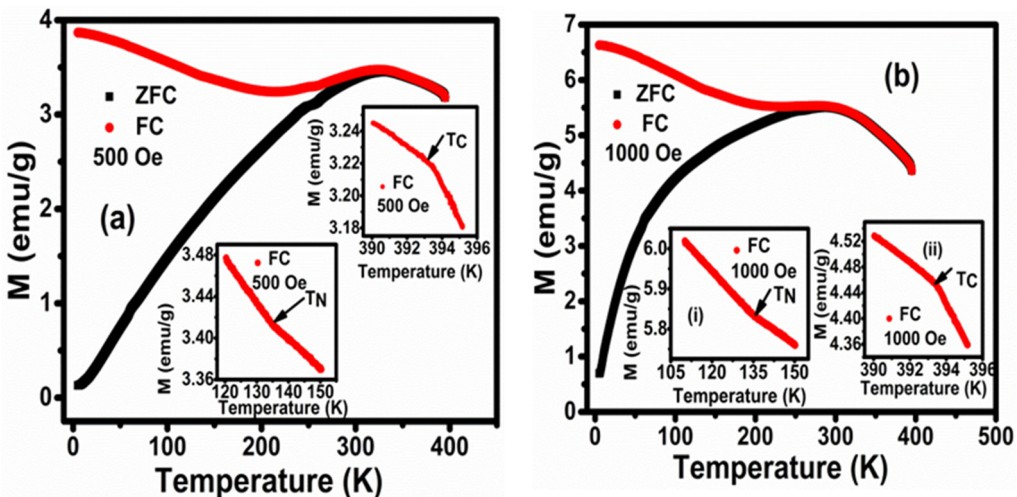

**Figure 2.** Temperature dependence of magnetization under ZFC and FC conditions of the PC10 composite at (**a**) H = 500 Oe and (**b**) H = 1000 Oe, the inset shows a magnified view of the ZFC curve at selected temperature regions.

The M-T study in this compound revealed that the magnetic property was arising from the contribution of the constituent phases PFN and CZFMO. To elucidate the ferromagnetic and antiferromagnetic/paramagnetic contributions to the magnetic properties, we tried to separate these contributions by fitting the M-H data. The M-H curves so obtained can be described by the expression [29]

$$M(H) = \left[ \frac{2M_{S(FM)}}{\pi} \tan^{-1}\left\{ \left( \frac{H \pm H_{ci}}{H_{ci}} \right) \tan \frac{\pi M_{r(FM)}}{2M_{S(FM)}} \right\} \right] + \chi H \tag{1}$$

where $M_{S(FM)}$, $M_{r(FM)}$, $H_{ci}$, $\chi$, and $H$ represent the saturation magnetization, remanent magnetization, coercivity, magnetic susceptibility, and applied magnetic fields. The first term of Equation (1) is for the ferromagnetic component, and the linear part (term outside square bracket) represents the antiferromagnetic/paramagnetic contribution. This equation was able to fit the M-H graph in the entire region, and hence, a good fit was observed between the experimental and the simulated data (shown in Figure 3a).

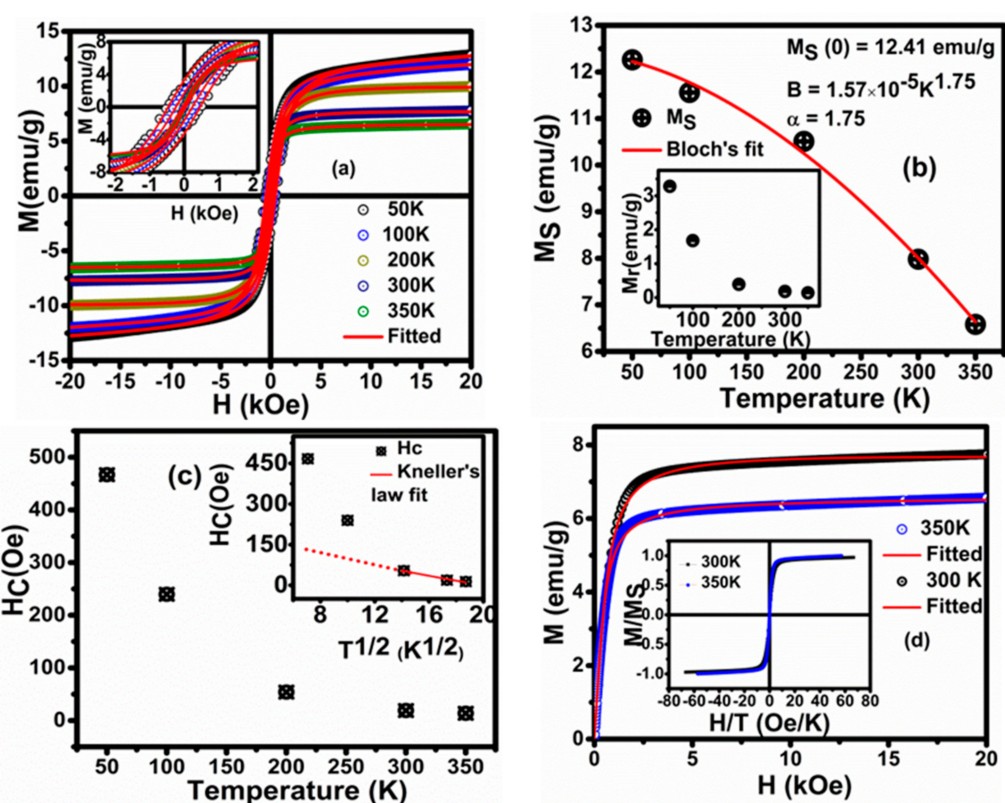

**Figure 3.** Variation of (**a**) M-H curves at different temperatures (50 K, 100 K, 200 K, 300 K, and 350 K). (**b**) Temperature variation of saturation magnetization ($M_S$) fitted with Equation (2). Inset shows the Mr vs. T plot. (**c**) $H_C$ vs. T plot. Inset shows the Kneller's law fit in the temperature region 200 K–350 K. (**d**) Fitting of the modified Langevin function. Inset shows the M/M$_S$ vs. H/T curve for the composite at 300 K and 350 K.

The saturation magnetization ($M_S$) value found from the fitting of M-H data decreased gradually with the increase in temperature shown in Figure 3b. The temperature dependence of $M_S$ in a magnetic system is described by Bloch's law written as

$$M_S(T) = M_S(0)[1 - BT^\alpha] \tag{2}$$

where $M_S(0)$ = saturation magnetization at $T \rightarrow 0$, $B$ = Bloch's constant, and $\alpha$ depends on the size of the particle. The exponent $\alpha$ is 3/2 for a bulk magnetic system, where the gap induced in the dispersion relation of a spin-wave is zero [34]. The fitting of $M_S$ vs. $T$ gives $\alpha = 1.75$, which suggests it to be close to Bloch's law with a small contribution of nano nature, where the $\alpha$ value has been found to be 2. The inset of Figure 3b shows a smooth decrease in $M_r$ with the increase in temperature.

The $H_C$ value also increased gradually with the decrease in temperature, and below 200 K, its value increased sharply (shown in Figure 3c). The thermal behavior of $H_C$ can be described by Kneller's law [34] given by

$$H_C = H_{C_0}\left[1 - (T/T_B)^{1/2}\right] \tag{3}$$

Here, $H_{C_0}$ is the coercivity at 0 K, and $T_B$ is the superparamagnetic Blocking temperature. The gradual increase in $H_C$ up to 200 K was may be due to the dynamical blocking of moments corresponds to the intrinsic anisotropy barrier [34]. The sharp enhancement of $H_C$ below 200 K could be arising due to a frozen spin-glass-like state, where more energy was required to flip the frozen surface spins, a consequently sharp rise in $H_C$ [34,35]. The Kneller's law fitted data (in the temperature range 200 K–350 K) is shown in the inset of Figure 3c, which gave the fitting parameter of $H_{C_0}$ = 332 Oe and $T_B$ = 380 K, which was also

observed from the other observation. The linear increase in $H_C$ at this temperature might be due to the thermal blocking of the spin moment arising from the intrinsic anisotropy barrier [34,35].

From the graph, one can observe very low values of coercivity and remanent magnetization around and above 300 K, which could be an indication of the existence of superparamagnetic behavior in the composite. Because the essential condition for superparamagnetic behavior in the material was generally based on two criteria, (i) there was no hysteresis (almost zero coercivity) in the M-H loop, and (ii) the curve of $M/M_S$ vs. $H/T$ (reduced magnetization curve) merged into a universal curve [36,37]. To probe this feature, we plotted $M/M_S$ vs. $H/T$ at 300 K and 350 K (shown in the inset of Figure 3d), i.e., in the superparamagnetic region. The overlapping of $M/M_S$ vs. $H/T$ at this temperature with almost zero loop opening was the manifestation of the superparamagnetic nature of the material.

In this scenario, the modified Langevin function can be used to describe the initial magnetization curves. The modified Langevin function can be expressed as [36,37]

$$M = M_S \left[ \coth(\mu_{eff}H/k_BT) - k_BT/\mu_{eff}H \right] + \chi_P H \tag{4}$$

where $M_S$ represents the saturation magnetization, $\mu_{eff}$ is the effective magnetic moment, $k_B$ is the Boltzmann constant, T is the absolute temperature, and $\chi_P$ is the paramagnetic component. A good fit between the experimental and theoretical data has been observed. The $M_S$ values at 300 K and 350 K were found to be 7.92 emu/g, 6.54 emu/g, respectively. The paramagnetic contributions ($\chi_P$) around this temperature region were $6.52 \times 10^{-6}$ (300 K) to $1.90 \times 10^{-6}$ (350 K).

### 3.3. Dielectric Study

Figure 4a depicts the temperature dependence dielectric constant of the PC10 composite for various frequencies. The value of the dielectric constant decreased with the increase in frequency, which is a typical characteristic of polar dielectric material. The dielectric constant value increased gradually with the increase in temperature attained a maximum of around 365 K, irrespective of all frequencies. After this maximum, its value decreased monotonically with a further increase in temperature. This observed maximum at 363 K corresponded to the ferroelectric-to-paraelectric transition ($T_C$) temperature. The composite showed a diffuse type of ferroelectric transition, and the value of the dielectric constant was less compared to the parent ($\Phi = 0.0$) PFN compound. The decrease in the dielectric constant with the addition of CZFMO could be due to the influence of the low-resistive cubic phase. The temperature dependence dielectric loss spectra also showed changes (anomaly) across $T_C$. However, at higher temperatures, the dielectric loss increased rapidly with the increase in temperatures.

In order to get information about the order of transition, the inverse of relative permittivity with temperature was plotted (shown in Figure 4b). The data at 100 kHz was selected to plot this graph in order to rule out the contribution from space charge polarization, which was dominant at the low-frequency region. According to the Landau–Devonshire theory of ferroelectric phase transition, the reciprocal of the dielectric permittivity for the first-order ferroelectric-to-paraelectric phase transition can be written as [38]

$$\frac{1}{\varepsilon} = 8\beta(T - T_C) + \frac{3\gamma^2}{4\delta}, T \to T_C^- \text{ For } (T < T_C) \tag{5}$$

$$\frac{1}{\varepsilon} = \beta(T - T_C) + \frac{3\gamma^2}{16\delta}, T \to T_C^+ \text{ For } (T > T_C) \tag{6}$$

and for the second-order ferroelectric-to-paraelectric phase transition, the reciprocal of dielectric permittivity can be expressed as

$$\frac{1}{\varepsilon} = 2\beta(T - T_C) \text{ For } (T < T_C) \tag{7}$$

$$\frac{1}{\varepsilon} = \beta(T - T_C) \text{ For } (T > T_C) \tag{8}$$

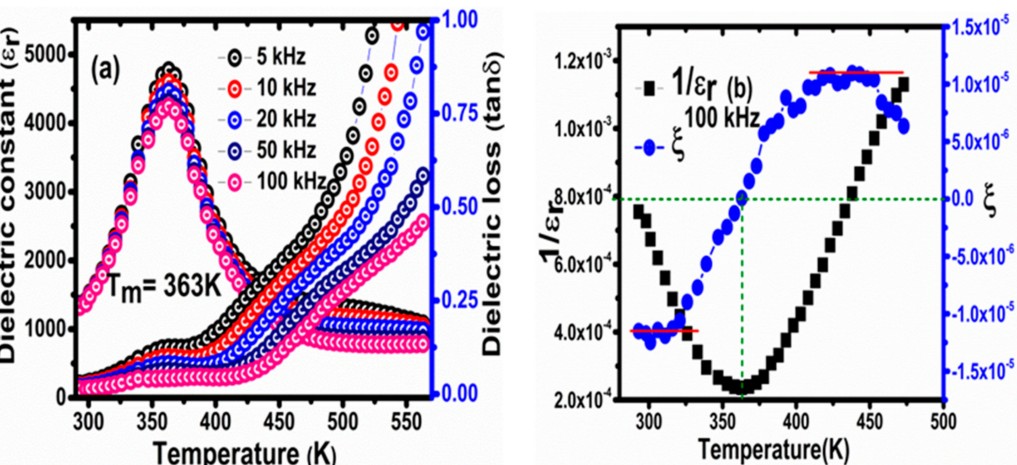

**Figure 4.** (**a**) Dielectric constant vs. temperature ($\varepsilon_r$) and dielectric loss (tan $\delta$) vs. temperature plot of PC10 for some selected frequencies. (**b**) Temperature dependence reciprocal of the dielectric constant ($\frac{1}{\varepsilon}$) and its temperature derivative ($\xi$) at 100 kHz.

Here, $\beta, \gamma$, and $\delta$ are the coefficients in the free energy expressions, and $T_C$ is the ferroelectric-to-paraelectric transition temperature. Hence, for a second-order phase transition, the quantity $\xi = \frac{\partial}{\partial T}(\frac{1}{\varepsilon})$ changed from $\beta$ for $T > T_C$ to $2\beta$ for $T < T_C$. This ratio of 2:1 is rarely observed for ferroelectric and ferroelastic, as reported by Ettandonea [39]; if it occurs, then it clearly signifies a second-order mean-field transition.

To validate the order of transition in the present system, we plotted $\frac{1}{\varepsilon}$ vs. T and $\xi$ vs. T in Figure 4b. The $\frac{1}{\varepsilon}$ vs. T plot showed different slopes for $T < T_C$ and $T > T_C$. The $\xi = \frac{\partial}{\partial T}(\frac{1}{\varepsilon})$ parameter was argued to give an idea about the order of transition along with the transition temperature, the temperature, at which $\xi = 0$ corresponds to ferroelectric-to-paraelectric phase transition. In the present compound, $T_C$ was calculated to be 365 K, the value of $\xi$ remained temperature independent below 330 K (in the ferroelectric region), also it remains constant above 420 K (in the paraelectric region). At the neighboring temperature of $T_C$, the value of $\xi = 2.80 \times 10^{-6}$ (373 K) changed to $\xi = -1.29 \times 10^{-6}$ at 357 K. The ratio of $\xi$ value above and below the transition was $\approx -2.17$, suggesting a second-order ferroelectric-to-paraelectric phase transition [40]. Therefore, it could be concluded that the PFN phase in the composite underwent a second-order ferroelectric-to-paraelectric phase transition, which remains unaffected by the addition of the CZFMO phase with a slight change in $T_C$ temperature.

### 3.4. Magneto-Dielectric Study

The presence of ferroelectric and ferromagnetic ordering in this system suggested a simultaneous existence of ferroelectricity and ferromagnetism, i.e., multiferroic behavior at RT [29]. In order to probe the existence of magneto-electric coupling in this composite, the magneto-dielectric properties were studied in detail. The magneto-dielectric effect in a system can be analyzed through (a) changes in the dielectric permittivity by applying a magnetic field and/or (b) indication of an anomaly near the magnetic phase transition temperature in the temperature-dependent dielectric spectra [41]. So, in order to investigate the presence of magneto-dielectric properties in the present system, we measured

electrical parameters such as capacitance and impedance with frequency for various applied magnetic fields. The change in the dielectric permittivity and impedance by the application of a magnetic field can be quantified by means of magnetocapacitance (MC%) and magnetoimpedance (MI%). The MC% and MI% were calculated using the expressions reported elsewhere [41]. The variations of MC% and MI% with applied magnetic fields (H; in increasing and withdrawing magnetic fields) are shown in Figure 5a,b at 10 kHz. Here, we observed a clear hysteresis in MC% vs. H and MI% vs. H while increasing and withdrawing H. The MC% and MI% for H = 1.25 T at 10 kHz were found to be 0.18% and 0.17% respectively. The change in dielectric parameters was small due to an averaged signal of the capacitive measurement over a relatively large electrode. However, the intrinsic coupling might be larger than the obtained value. Further, the appearance of hysteresis might be applicable in passive filters, where the zero-field dielectric permittivity of the systems depends on the magnetic hysteresis of the specimen and thus suitable for device fabrications [42,43]. The magnetocapacitance arose due to the magnetostriction and piezo-electricity through strain transfer at the interface between the individual phases. In the high-frequency region, the contribution from space charge polarization could be excluded, and the intrinsic nature of magneto-dielectric properties could be realized. In this composite, the dielectric loss (tanδ) value was very small, and, further, the intrinsic magnetic susceptibility $\chi_{ME}$ could be calculated as described by Jang et al. [44], which is given by

$$\frac{\varepsilon(H) - \varepsilon(0)}{\varepsilon(0)} = \frac{\chi_{ME}}{E_0} \times H \qquad (9)$$

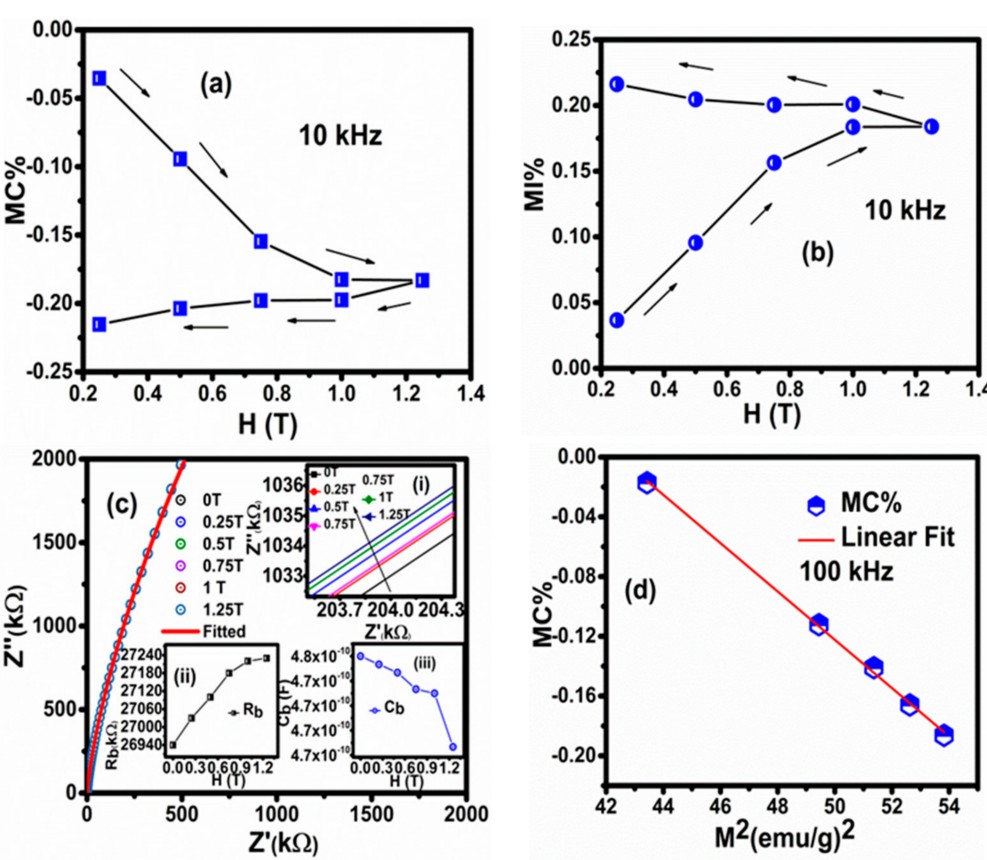

**Figure 5.** Variation of (**a**) MC% and (**b**) MI% with the magnetic field. (**c**) Nyquist plot for various magnetic fields, inset (**i**) is the effect of the magnetic field Nyquist plot (**ii**) bulk resistance (R$_b$) vs. the magnetic field (**iii**) bulk capacitance (C$_b$) vs. the magnetic field. (**d**) MC% vs. M$^2$.

$\varepsilon(H)$, $\varepsilon(0)$ are the values of dielectric permittivity in the presence and absence of the magnetic field. $E_0$ is the applied electric field excitation (excitation voltage 1 V, thickness

of the sample = 1 mm). The $\chi_{ME}$ value for the present system was calculated to be 0.14 mVcm$^{-1}$Oe$^{-1}$, which is comparable to the typical multiferroic composites BaTiO$_3$-CoFe$_2$O$_4$ for probing frequencies higher than their conductivity cutoff [45].

Generally, in a multiferroic system (in polycrystalline form), the magnetocapacitance may have contributions from some external factors, such as grain boundaries, material electrode interface, etc., of different resistivity. Further, a mismatch of the work function between the electrode and dielectric material at the interface results in a strong change of dielectric parameters by the application of magnetic fields. The extrinsic contribution to the magneto-dielectric effect can be separated from the magnetoimpedance measurement, and always the contribution from bulk capacitance is intrinsic in nature. The Nyquist plot for the composite PC10 for various applied magnetic fields is shown in Figure 5c. The experimentally-obtained data are fitted by considering the (R$_b$C$_b$Q) equivalent circuit. (R$_b$C$_b$Q) represents the parallel combination of bulk resistance (R$_b$), bulk capacitance (C$_b$), and the constant phase element (CPE, Q). The justification for proposing this circuit was described in our previous article [46]. From the fitting, the obtained parameters bulk resistance (R$_b$) and bulk capacitance (C$_b$) with the magnetic field are shown in the inset of Figure 5c. The R$_b$ value was found to be increased with the increase in the magnetic field, whereas the C$_b$ value decreased with the increase in the magnetic field. The change of C$_b$ with the magnetic field suggested the existence of intrinsic magneto-dielectric behavior in the present compound, as the bulk capacitance was always free from the interfacial effects.

In a ferro electromagnet, the thermodynamic potential $\psi$ can be written in the form [47]

$$\psi = \psi_0 + \alpha_1 P^2 + \frac{\beta_1}{2}P^4 - PE + \alpha_1' M^2 + \frac{\beta_1'}{2}M^4 - MH + \gamma_1 P^2 M^2 \tag{10}$$

where P and M are the polarization and magnetization, respectively. The term $\alpha_1$, $\beta_1$, $\gamma_1$ … are the coupling coefficients. The term containing biquadratic coupling $P^2M^2$ is always allowed from the symmetry point of view [47]. It has been well known that $\Delta\varepsilon_r \sim \gamma_1 M^2$, $\gamma_1$ may possess positive or negative value [47]. The MC% vs. $M^2$ shows a linear variation (as shown in Figure 5d), and the absolute value of $\gamma_1$ was found to be $\sim 1.5 \times 10^{-2}$ (emu/g)$^{-2}$. The linear variation of MC% vs. $M^2$ told that the magneto-dielectric effect in the PC10 composite arose from the biquadratic coupling term $\gamma_1 P^2M^2$ in the expression of thermodynamic potential (Equation 10). The biquadratic coupling was the result of the piezoelectric and magnetostrictive effect of the constituent phases that led to the square of their respective order parameters.

### 3.5. Converse Magneto-Electric Effect (CME)

The electric field control of magnetic ordering, also known as converse magneto-electric effect (CME), is studied by measuring the M-H loop after electrical poling of the sample (at 20 kV/cm) at RT. A reduction in the magnitude of magnetic parameters, such as $M_S$, $M_r$, and $H_C$, was observed for the poled sample, as shown in Figure 6. The $M_S$, $M_r$, and $H_C$ were reduced by 24.7%, 58.3%, and 38.1%, respectively, after electrical poling on the sample. This may be attributed to the occurrence of elongation of the ferroelectric phase along the applied field direction when an external electric field was applied to the composite. This elongation generated compressive stress along the applied electric field direction, which caused tensile stress along the magnetic field direction in the magnetic phase since the generated compressive stress was perpendicular to the magnetic field direction. Further, the magnetostrictive phase CZFMO underwent negative magnetostriction, according to the earlier study. So, the magnetization decreased along the magnetic field direction, which was also the measuring direction of magnetization [18]. So the changes in the magnetic behavior due to electrical poling indicated the existence of CME. Therefore, PC10 exhibited CME at RT and might be applicable in multifunctional devices, which could be operated at RT.

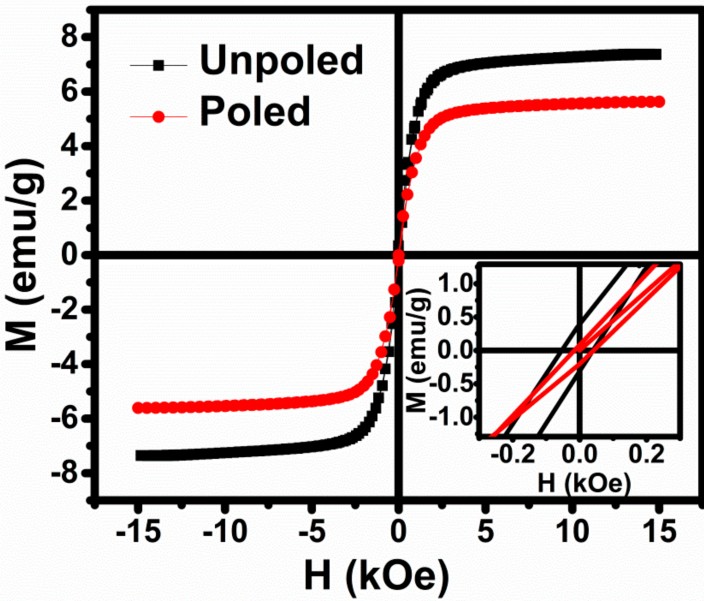

**Figure 6.** M vs. H data of PC10 composite taken before and after electric field poling; the inset shows the effect poling on the low magnetic field region.

## 4. Conclusions

The multiferroic composite PC10 was prepared using a hybrid synthesis technique. The presence of both the parental phases, i.e., PFN and CZFMO in the XRD patterns, suggested the formation of composites. The dispersion of the CZFMO phase in the matrix of the PFN phase revealed the successful formation of the particulate composite. The M-T measurement of the composite gives an idea about the magnetic ordering of the PFN as well as the CZFMO phase. M-H data taken at different temperatures suggested the existence of superparamagnetic behavior in the composite above 300 K. Temperature-dependent dielectric measurement gave a second-order ferroelectric-to-paraelectric phase transition of the PFN phase. The magneto-dielectric measurement indicated a change of the dielectric parameters and, hence, suggested the presence of magneto-dielectric behavior. The magnetic parameters changed by electric field poling of the sample, which further suggested the existence of strong magneto-electric coupling behavior. The RT magneto-electric behavior of the present composite may be useful in potential device applications.

**Author Contributions:** K.B.: Formal analysis, investigation, methodology, writing—original draft, S.D. (Smaranika Dash): Data curation and validation, S.D. (Sita Dugu): Data curation and validation, D.K.P. (Dhiren K. Pradhan): Data curation, validation, writing—review and editing, A.K.S.: Data curation, validation, writing—review and editing, P.N.V.: Data curation, validation, writing—review and editing, R.S.K.: Data curation, validation, writing—review and editing, D.K.P. (Dillip K. Pradhan): Conceptualization, supervision, validation, writing—review and editing. All authors have read and agreed to the published version of the manuscript.

**Funding:** This research received no external funding.

**Acknowledgments:** The authors (K.B. and D.K.P.) acknowledge the UGC DAE, CSR Mumbai Centre (project code: UDCSR/MUM/AO/CSR-M-198) for financial support. R.S.K. acknowledges the DoD Project #FA9550-20-1-0064.

**Conflicts of Interest:** The authors declare no conflict of interests.

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
