# Peer review of "Investigation of the Phase Transitions and Magneto-Electric Response in the 0.9(PbFe0.5Nb0.5)O3-0.1Co0.6Zn0.4Fe1.7Mn0.3O4 Particulate Composite"

_jcs, doi:10.3390/jcs5070165_

Round 1
Reviewer 1 Report
The subject of the manuscript is very interesting. The authors have conducted a concerning composite which is the combination of multiferroic (PbFe0.5Nb0.5O3, PFN) as matrix and magnetostrictive (Co0.6Zn0.4Fe1.7Mn0.3O4, CZFMO) material as the dispersive phase. In my opinion, this is an essential topic. The authors performed tests that confirmed the formation of composite having both perovskite PFN and magnetostrictive CZFMO phases. The title fully describes the subject matter of the article. I want to press the point that the authors of the manuscript describe the information, which that may be of use to other researchers when working on the investigation of phase transitions and magneto-electric response in multiferroic composites.
The presented data are reliable and useful. However, the paper needs some improvement only after which it can be published.
- In my opinion, the authors of the manuscript should necessarily more refer to the newer publications from 2019-2021 in the "Introduction" section. The authors of the manuscript refer to their work to publications from 2008-2017 and one manuscript from 2019 and one article from 2021. It seems to me that there are many curious and interesting works about multiferroic composites with an enhanced magneto-electric coefficient that have been published in recent years, and which could be referred to in the "Introduction" part of the manuscript.
- What was the counting time during the XRD measurement?
- In my opinion, the SEM investigation procedure is poorly described. The reviewer recommends adding information about the SEM device, preparation procedure of the samples for the investigation, etc.
- The quality of the pictures is very poor, please change the resolution of the pictures to a higher one.
After implementing minor corrections, I recommend the article for publication.
Author Response
Title: Investigation of Phase Transitions and Magneto-electric Response in 0.9(PbFe0.5Nb0.5)O3-0.1Co0.6Zn0.4Fe1.7Mn0.3O4 Particulate Composite
Authors: Krishnamayee Bhoi, S. Dash, S. Dugu, Dhiren K. Pradhan, A. K. Singh, P. N. Vishwakarma, R. S. Katiyar, Dillip K. Pradhan
Response to the Reviewer’s Comment
We are thankful to the reviewers for their positive appraisal and valuable suggestions to improve the quality of the manuscript. The manuscript has been revised based on the reviewers’ comments. All those imprecisions and mistakes are corrected, and we also have carefully gone through the manuscript for improving the quality of the manuscript. We believe we have addressed his/her concerns below. The blue colored sentences in the revised manuscript denote added/modified text that address the reviewer’s suggestions. Detailed reply to reviewers’ comments is as follows:
Response to Reviewer-1 Comments
Comment 1: In my opinion, the authors of the manuscript should necessarily more refer to the newer publications from 2019-2021 in the "Introduction" section. The authors of the manuscript refer to their work to publications from 2008-2017 and one manuscript from 2019 and one article from 2021. It seems to me that there are many curious and interesting works about multiferroic composites with an enhanced magneto-electric coefficient that have been published in recent years, and which could be referred to in the "Introduction" part of the manuscript.
Response: As per reviewer’s suggestion we have included and discussed few recent works on particulate multiferroic composite between the years 2019-2021 in the "Introduction" section.
This is included on P. 2-3 of the revised manuscript. The references are in accordance with the revised manuscript.
Comment 2: What was the counting time during the XRD measurement?
Response: The XRD data were collected in a wide range of Bragg’s angle 2θ (20°≤2θ≤70°) with the step size of 0.02° at a scan speed of 1 degree/minute.
This is included in p.4 of the revised manuscript.
Comment 3: In my opinion, the SEM investigation procedure is poorly described. The reviewer recommends adding information about the SEM device, preparation procedure of the samples for the investigation, etc.
Response:
The surface morphological features were studied using an automated Scanning Electron Microscope (SEM-JEOL-JSM 6480 LV). The sintered pellet was being gold coated under vacuum prior to the SEM investigations. The SEM images were captured at various magnifications.
This is included in p.4 of the revised manuscript.
Comment 4: The quality of the pictures is very poor, please change the resolution of the pictures to a higher one.
Response: As per reviewer’s suggestion the figures have been modified.

Reviewer 2 Report
The quality of the drawings should be improved - the resolution (the drawings are blurry, you can see that they are enlarged from a small size), the thickness of the lines and the size of the measurement points - to increase the readability.
Author Response
Title: Investigation of Phase Transitions and Magneto-electric Response in 0.9(PbFe0.5Nb0.5)O3-0.1Co0.6Zn0.4Fe1.7Mn0.3O4 Particulate Composite
Authors: Krishnamayee Bhoi, S. Dash, S. Dugu, Dhiren K. Pradhan, A. K. Singh, P. N. Vishwakarma, R. S. Katiyar, Dillip K. Pradhan
Response to the Reviewer’s Comment
We are thankful to the reviewers for their positive appraisal and valuable suggestions to improve the quality of the manuscript. The manuscript has been revised based on the reviewers’ comments. All those imprecisions and mistakes are corrected, and we also have carefully gone through the manuscript for improving the quality of the manuscript. We believe we have addressed his/her concerns below. The blue colored sentences in the revised manuscript denote added/modified text that address the reviewer’s suggestions. Detailed reply to reviewers’ comments is as follows:
Response to Reviewer-2 Comment
Comment:
The quality of the drawings should be improved - the resolution (the drawings are blurry, you can see that they are enlarged from a small size), the thickness of the lines and the size of the measurement points - to increase the readability.
Response: As per reviewer’s suggestion the figures have been modified.
